# "Lamp and Candle": Classical Chinese Imagery in Taixu's Poetry

Xiaoxiao Xu

Faculty of Arts and Philosophy, Ghent University, 9000 Ghent, Belgium; xiaoxiao.xu@ugent.be

**Abstract:** Taixu 太虛 (1890–1947), a prominent figure in modern Chinese Buddhism, produced a voluminous collection of poetry abounding with diverse classical Chinese images. Notably, the "lamp and candle" (*dengzhu* 燈燭) holds great significance, reflecting Taixu's personal affinity with this imagery and an intimate connection to classical Chinese poetry. Acting as a potent Buddhist metaphor, it encapsulates multifaceted sentiments while also intertwining with other evocative images, such as the boat, the moon, and falling leaves. Symbolizing Taixu's unwavering spirit, it represents his profound dedication to his craft. This article explores Taixu's literary achievements as a poet by focusing on his adept utilization of "lamp and candle" imagery, complementing the study of his multifaceted and intricate identities. This detailed examination offers novel insights into Chinese literature and Buddhist studies, highlighting the interplay between spiritual practice and artistic expression.

**Keywords:** classical Chinese imagery; poetry; Taixu; lamp and candle

## 1. Introduction

During the late nineteenth and early twentieth centuries, China experienced a notable revival of Buddhism, which led, in turn, to the advancement of Buddhist literature (Tan 2010).[1] Taixu, a prominent scholar who played a key role in organizing and promoting this Buddhist movement, had a deep connection with classical Chinese literature, particularly poetry. Born into a literary family, he was raised by his grandmother, Zhou Lixiu 周理修, a knowledgeable practitioner of Buddhism and Daoism who specialized in poetry (TDQ, vol. XXXI, pp. 156–57). Moreover, he received guidance from his uncle, Zhang Zizang 張子綱, a talented literatus who was well versed in Chinese literature (ibid., pp. 159–60). This upbringing and educational background deeply influenced Taixu's early involvement with poetry, which was further nurtured by his exceptional intellect and prodigious memory (ibid., p. 159). Remarkably, he started learning and reciting poetry at the tender age of five, with a specific focus on the Tang Dynasty (618–907), which he felt represented the zenith of classical Chinese poetry (ibid., p. 159).[2] Throughout his formative years, he continued to hone his own poetic skills by collaborating with mentors, companions, disciples, and students.[3] Notably, in 1916, he achieved a significant milestone by publishing his inaugural poetry collection—*Meian shilu* 昧盦詩錄 (*The Poetry Collection of Meian*)—which cemented his reputation as a recognized poet (ibid., p. 197).[4] Thereafter, he continued to dedicate himself to his craft until his passing in 1947 (TDN, p. 346).

As a prolific poet for more than three decades, Taixu produced a significant corpus of over a thousand poems. These works, which include collaborations as well as individual compositions, are compiled in the "Shicun 詩存" (Poetry Collection) chapter of the *Taixu dashi quanshu* 太虛大師全書 (*Collected Works of Master Taixu*), a comprehensive anthology edited primarily by Yinshun 印順 (1906–2005),[5] one of Taixu's most distinguished disciples. A number of additional verses appear in the *Taixu dashi nianpu* 太虛大師年譜 (*Chronological Biography of Taixu*), also edited by Yinshun, and *Taixu zizhuan* 太虛自傳 (*Taixu's Autobiography*). All of these poems serve as invaluable resources for exploring Taixu's philosophical beliefs, personal emotions, and remarkable accomplishments.

However, despite extensive studies on various aspects of Taixu's life, research has often focused on his identities as a reformer, political activist, and educator, while neglecting his identity as a poet.[6] In this sense, his poetry has been overlooked as a valuable resource for understanding his multifaceted personality. Interpreting Taixu's poetry from both Buddhist and literary perspectives will undoubtedly add crucial information on the role he played in Chinese society.

Taixu's poetry is replete with classical Chinese imagery, with the "lamp and candle" especially prevalent and prominent. As essential sources of light, lamps and candles have held deep symbolic significance in Chinese literature since ancient times, particularly within the realm of poetry.[7] Furthermore, this imagery has experienced a resurgence in popularity among modern poet-monks.[8] Taixu, following in the footsteps of generations of Chinese literati, deftly employed "lamp and candle" imagery in over sixty of his poems, establishing himself as one of the most prolific users of this motif. In this way, he not only conveyed his personal perspectives and emotions but also showcased his unique artistic sensibilities.

This article will explore a series of questions relating to this imagery:

- Why is the "lamp and candle" so prominent in Taixu's poetry?
- What symbolic meaning does this imagery hold as a Buddhist metaphor?
- Which personal sentiments are expressed through Taixu's use of this imagery?
- How does this imagery serve as a conduit for transmitting Taixu's personal emotions and spiritual experiences?

Scrutinizing Taixu's adept utilization of "lamp and candle" imagery in his poetic compositions provides unprecedented insights into how he reinforced his identity as a poet by fusing artistic expression with religious devotion.

The article is founded upon three primary objectives:

- It presents an uncharted examination of Taixu's accomplishments as a poet-monk, serving as a complementary study to his multifaceted and intricate identities.
- It illuminates Taixu's extensive collection of poetry, which has received scant attention from both Buddhist and literary scholars.
- It innovatively analyses the metaphorical image of the "lamp and candle", which held profound influence over Taixu's life and monastic vocation, yet has remained largely overlooked in the comprehensive inquiries into various facets of his life.

## 2. The Significance of "Lamp and Candle" Imagery in Taixu's Poetry

Before embarking on a thorough exploration of the philosophical and literary aspects of "lamp and candle" imagery, it is imperative to address its significance in Taixu's poetry. Although he possessed a personal fondness for lamps and candles (which served as the main sources of illumination in his daily life), the transformation of these utilitarian objects into weighty poetic imagery was far from straightforward. Rather, it was a complex process that reflected the poet's upbringing, education, and subsequent life experiences.

Taixu developed a profound affinity for lamps and candles from a very early age. For example, he recalled experiencing the warm radiance of an oil lamp for the first time while residing with his grandmother—a devout Buddhist—in the Dayin Nunnery (Dayin an 大隱庵) at the age of five (TDQ, vol. XXXI, p. 156).[9] This initial encounter with a lamp's soothing glow left an indelible imprint upon the young Taixu, as he explains in his autobiography:

> 我最早的意識和想像，是庵內觀音龕前的琉璃燈；有一次看著外婆把燈放下來，添了油，燃了火，又扯上去，注視得非常明晰深刻。同時，並想像屋樑下懸有一個什麼靈活的東西在牽動著，而各種知識記憶乃從此萌芽了。

> My earliest consciousness and imagination revolved around the glass lamp that stood before the Guanyin (Avalokiteśvara) altar in the nunnery. On one occasion, I watched as my grandmother placed the lamp down, replenished its oil, kindled the flame, and raised it once more, observing the entire process with remarkable

clarity and profundity. Concurrently, I imagined an agile presence suspended from the roof beam, a moment from which various memories and knowledge sprouted forth.[10]

This mysterious and romantic experience holds a significant place in Taixu's earliest childhood memories. The illuminating glass lamp, symbolizing his curiosity and thirst for life, became an indelible image deeply ingrained in his mind. While Taixu refrained from explicitly elucidating this "agile presence", we may infer that it played a key role in his enlightenment and the development of his craft. Indeed, it is possible to trace his numerous depictions of "lamp and candle" imagery within religious contexts, including temples, churches, and other sacred spaces, to this early, wondrous encounter. For instance, more than three decades later, in October 1928, during a visit to the Sacred Heart Basilica in Paris,[11] his attention was immediately drawn to the flickering candles:[12]

高處巋然古教堂，紛陳像設燭騰光。

High above, the ancient church majestically stood, Statues and candles arranged, a dazzling flood.[13]

Taixu was clearly captivated by the sight of candlelight within temples and grand churches, and this is certainly a plausible explanation for his repeated use of "lamp and candle" imagery both during and after his travels. However, he was also able to detach it from any religious context and give it a more universal essence. This process was facilitated by the advent of the "electric lamp" (*diandeng* 電燈), which revolutionized traditional "lamp and candle" imagery.

The electric lamp made its first appearance in Beijing's Forbidden City (Zijin Cheng 紫禁城) in 1888,[14] just two years before Taixu's birth. Over the next six decades, his life unfolded amidst a gradual proliferation of electric lighting. During his youth and early adulthood, prior to a period of seclusion on Putuo Mountain (Putuo shan普陀山) in 1917,[15] the "lamp and candle" imagery in his work related exclusively to oil lamps and candles. However, following the conclusion of his retreat, Taixu's exposure to electric lighting increased significantly as he ventured abroad and started to spend considerable amounts of time in major, modern cities. For example, a burgeoning fascination with the warm glow of electric lamps is evident in a poem he composed en route to Japan in the autumn of 1917:

驚起鴛鴦眠不定，粲然微笑電燈前。

Stirred awake like the restless mandarin ducks, [I] sparkle with a smile before the electric lamp.[16]

Taixu had dreamed of this journey for some time, as he had long hoped to study the amalgamation of Buddhism and Western philosophy in a country where East met West (TDQ, vol. XXXI, p. 288). However, on board the ship, at midnight, amidst autumnal winds and rain, his excitement was tinged with trepidation. Restless and unable to sleep, he finally found solace in the light from an electric lamp. His admiration for the bulb's comforting radiance imbued the "lamp and candle" imagery with a modern and rational essence that distinguished it from classical Chinese images, such as mandarin ducks.

Five years later, in 1922, Taixu visited Yingjiang Temple (Yingjiang si 迎江寺), a renowned Chan monastery in Anqing安慶, and subsequently composed a poem dedicated to its Zhenfeng Pagoda (Zhenfeng ta 振風塔):[17]

雲樹風帆明遠近，電燈高照大江南。

Clouds, trees, wind, sails, far and near, Electric light high shines, over the great river south.[18]

Here, a number of classical Chinese elements—clouds, trees, wind, and sails—epitomize the natural beauty that envelops the temple, while the electric light infuses the scene with modern brilliance and dynamism. The convergence of these natural and artificial images

achieves a level of harmonization unparalleled in classical Chinese poetry, facilitated by the poet's enduring reverence for "lamp and candle" imagery.

Taixu's skillful incorporation of this imagery into his work may be attributed, at least in part, to his educational background and, particularly, his study of Tang poetry. His formal introduction to the genre began in *mengguan* 蒙館,[19] where his knowledgeable uncle played a pivotal role (TDQ, vol. XXXI, p. 159). Thereafter, as his schooling continued through interactions with various masters and friends in monasteries, as well as self-study, his respect for his Tang predecessors remained steadfast (TDQ, vol. XXXI, pp. 169–74). Many of these poets frequently employed "lamp and candle" imagery in their work (TDQ, vol. XXXII, p. 414). For example, in the *Quan Tangshi* 全唐詩 (*Complete Collection of Tang Dynasty Poetry*),[20] lamps are mentioned 1,563 times, while candles appear 986 times (Fu 2007, p. 234). One of Taixu's favorite poets, Bai Juyi 白居易 (772–846),[21] used such imagery no fewer than 185 times in the course of his career (ibid.), including in the following couplet:

香火一爐燈一盞，白頭夜禮佛名經。

A furnace of incense and a lamp in sight, At night with white hair, [I] pay my respects and recite.[22]

While the faint incense flames may soon fade, the light of the lamp will endure, enabling Bai Juyi, a devout Buddhist, to continue to read the scriptures and meditate. Hence, the lamp is far more than a simple source of illumination; it also lights the path to Buddhist wisdom. More than a millennium later, Taixu would echo this sentiment in his own poetry by utilizing similar imagery to convey spiritual insight and enlightenment.

### 3. The Mind Lamp: A Buddhist Metaphor

Taixu uses the "mind lamp" (*xindeng* 心燈) metaphor to draw parallels between the mind's cognitive functions and lamplight. Specifically, this metaphor, which is deeply rooted in Buddhist tradition,[23] finds expression in two distinct dimensions of his work: the embodiment of Buddhist wisdom and the dissemination of the Buddhist Dharma. He first employed it in the spring of 1913, when writing an article in which he heralded the establishment of a new Buddhist association:[24]

吾人既稱佛子，不可不於佛法中自覓安心立命之地，以紹佛之心燈於不絕。

As practitioners of Buddhism, it is essential [for us] to seek within the teachings of Buddhism a place of inner peace and existential grounding, so that [we may] perpetuate the illuminating light of the Buddha's mind lamp.[25]

This elucidation accentuates the importance of studying Buddhist teachings as a means to illuminate the mind—analogous to using a lamp to dispel darkness. Taixu reiterates this message numerous times in his poetry. By drawing parallels between the lamp and the enlightenment that results from the Buddha's teachings, he underscores the former's significance as a symbol of illumination and spiritual awakening:

燈比佛果上之恩德，以燈能破闇，而佛之大悲，亦能破除眾生癡闇。

The illumination of the lamp symbolizes the meritorious compassion inherent in Buddha's fruition,[26] as the lamp dispels darkness, the profound compassion of the Buddha also dispels the ignorance of sentient beings.[27]

Hence, Taixu argues that studying Buddhism enables ordinary individuals to cultivate their mind lamps. Another text reiterates this point by highlighting the transformative effect of Buddhist enlightenment:

蓋漫漫生死長夜，佛為明燈；今既不信佛燈，是即失大利樂，如"盲人騎瞎馬，半夜臨深流"。

In the vast cycle of birth and death, which resembles a long night, the Buddha serves as an illuminating lamp. Those who fail to place their trust in the Buddha's

lamp relinquish immense benefits and joy, akin to a "blind person riding a blind horse, approaching a deep stream in the middle of the night."[28]

Along with many other metaphors in the Buddhist tradition (Fu 2007, p. 245), the lamp represents the inherent wisdom of Buddhism, while discovering the mind lamp symbolizes the process of acquiring that wisdom. Both feature prominently in Taixu's exchanges with fellow poets, as in the following example:

剔燈曾共展奇文，今日重來倍憶君。

We once shared the brilliance under the lamp's glow, Today, revisiting, memories of you (Huoxuan) multiply and grow.[29]

This couplet was written in response to a poem by Huoxuan 豁宣,[30] one of Taixu's closest friends, in 1916. When they met again after a long period apart, Taixu's most vivid memory was of them reading Buddhist scriptures together. In this context, "the brilliance under the lamp's glow" symbolizes not only the Buddhist wisdom they acquired prior to their separation but also their enduring bond of friendship. An identical sentiment is evident in a poem that Taixu wrote during an overnight stay at Lingyun Temple (Lingyun si 靈雲寺):[31]

主人寒夜來生客，一剔心燈耀古今。

On a wintry night, the host welcomes the guest, A flicker of the mind lamp illuminates the past and present.[32]

On this cold night, a lamp lit by the abbot for Taixu signifies the two monks' friendship and camaraderie. In addition, the mind lamp symbolizes the Buddhist wisdom that has persisted since ancient times, and perhaps the transmission of the Buddhist Dharma.

Taixu's poetry often incorporates the concept of the "endless lamp" (*wujin deng* 無盡燈)—an extension of the mind lamp metaphor that encompasses the transmission of the Buddhist Dharma. The interplay of light from an infinite number of lamps creates a space filled with radiant brilliance. Taixu clarifies this idea in a treatise entitled *Wo zhi zongjiao guan* 我之宗教觀 (*My Religious Views*), written in 1925:

所謂一室千燈，光光互遍，相涉無礙，不壞自相。

The so-called one room, thousand lamps scenario, where many lights permeate, intersect without hindrance, and do not diminish one another.[33]

According to Taixu, the Buddhist Dharma is transmitted through both promotion and education. For example:

修行證果，弘法利世，焰續佛燈明。

Engaging in spiritual practice to attain realization, Promoting the Dharma for the welfare of the world, The flame perpetuates the radiance of the Buddha's lamp.[34]

Therefore, those who study Buddhism transmit its teachings for the good of the world through their Buddhist practice, ensuring the perpetuation of the mind lamp. Taixu revisits this idea in his response to a poem by Shi Ji'an 施寄庵,[35] a Taiwanese 臺灣 journalist, following a Buddhist assembly (TDN, p. 59):

願將文字[36]有為法,[37] 傳作光明無盡燈！

May these written poems embody the conditioned phenomena, Transmitting as an infinite and radiant lamp![38]

Taixu is a strong advocate of disseminating the Buddhist Dharma through various means, ranging from inspirational poetry to the luminosity of countless mind lamps, but he also understands that education has a crucial role to play in this process:

新參僧教育，無盡佛明燈。

Through the education of novice monks, The infinite light of the Buddha shines endlessly.[39]

According to Taixu, such education is key to the reformation of Chinese Buddhism (TDQ, vol. XIX, p. 14).[40] The infinite light from devotees' mind lamps signifies the diffusion of the Buddhist Dharma throughout the world, filling Taixu with hope for the future of Buddhism in China. This is just one of the ways in which he uses the image of a warm lamp to symbolize his inner feelings.

### 4. Taixu's Personal Emotions in "Lamp and Candle" Imagery

Taixu's incorporation of metaphorical "lamp and candle" imagery in his poetry is intimately connected to his personal emotions. Within classical Chinese culture, such imagery often symbolizes the aspirations and hopes of individuals (Fu 2007, p. 248), as in the *Ben Cao Gang Mu* 本草綱目 (*Encyclopedia of Materia Medica*),[41] a renowned medical text:

燈花爆而百事喜。

When the flame of a lamp wick suddenly bursts forth, it signifies joyousness in every aspect.[42]

Similarly, the *Yi Wen Zhi* 藝文志 (*Treatise on Arts and Letters*),[43] the earliest comprehensive catalogue of books in China, includes descriptions of how people in ancient times would forecast the future based on the brightness of lamps and candles. As a classical Chinese literatus, Taixu was deeply influenced by the traditional significance of this imagery, with the warm lamps and glowing candles in his poetry frequently evoking a sense of tranquility and inner peace:

晚來偕向市場遊，燈火星羅車水流。

In the evening, [we] venture together to the bustling market, Where lanterns shimmer and carriages flow like a river.[44]

Taixu composed this couplet during a visit to Taipei in November 1917.[45] He found himself in a state of tranquility as he wandered through the bustling local market (signified by carriages flowing "like a river") under the illuminating glow of lanterns. It is important to note that this imagery symbolizes his sense of calm. Thus, rather than explicitly expressing his inner feelings, he skillfully allows a simple description of the market's lamplight to convey his emotions to the reader. The same technique is evident in a poem he wrote the following month in Kyoto:[46]

夜從書城搜盡殘，花海燈市才一瞥。

In the evening, [I] search for remnants of decay in the city of books, But caught only a fleeting glimpse of the sea of flowers and the bustling lamplit market.[47]

Here, the lamplight evokes Taixu's sense of inner peace as he catches an all-too-brief glimpse of natural beauty in the midst of the busy market. As a monk accustomed to a life of solitude, he clearly felt considerable unease when venturing into unfamiliar, public places, yet even the briefest glimpse of a lamp could relax him and inspire him to write poetry. That said, he preferred more serene environments, and particularly the tranquility of his own residence:

剪燈同夜話，高躋待明朝。

Engaging in late-night conversations under the dimly lit lamp, [We] ascend to higher grounds, awaiting the arrival of dawn.[48]

This poem records a conversation between Taixu and a visiting woodcutter on Jiuhua Mountain in 1929. Both the lengthy conversation itself and the elevated location reflect Taixu's excitement and joy in their dialogue. Once again, while the poet does not explicitly express his happiness, every word is infused with positivity, particularly in his depiction of the lamp that burns through the night.

Interestingly, there are more overt expressions of happiness in Taixu's later work, as in the following poem, which dates from the final few years of his life:

清談娓娓消初夜，喜有明燈耀案頭。

Engaging in a lively conversation, the first night dissipates gracefully, Delighted to have a bright lamp illuminating the desk.[49]

In his secluded private residence, which is usually characterized by tranquility and solitude, Taixu engages in conversations with friends that can continue throughout the night, using the term *xi* 喜 to convey the pleasure that these lamplit discussions provide.

Although Taixu often employs "lamp and candle" imagery as a metaphor for human behavior and emotions, this is not always the case. On occasion, he simply expresses the joy he feels when observing the play of light in a serene natural setting, far removed from any human activity:

上界繁星隔岸燈，湖天一碧萬光騰。

Across the celestial realm, stars twinkle, while lamps illuminate the distant shores, The lake and sky merge as one, a boundless expanse gleaming with myriad lights.[50]

Taixu was inspired to write this poem on a peaceful boating excursion on West Lake (*Xihu* 西湖) during his residence in Hangzhou in the summer of 1927 (TDN, pp. 154–55).[51] He focuses solely on the ethereal glow of the humble lamplight, which becomes a bridge between the vast expanse of the heavens above and the tranquil waters below. Through this artistic device, he effortlessly traverses the spatial constraints of the sky and the lake, crafting an intertwining realm of luminosity shaped by the radiance of stars and lamps. Once again, the reader senses Taixu's excitement at witnessing this enigmatic and resplendent landscape, even in the absence of any explicit emotional declaration. Furthermore, his exhilaration is accentuated by the insects that gather in the flickering glow of the lamplight:

雲中金粟[52]影，燈下草蟲啼。

In the realm of clouds, Buddha's radiance shines, Underneath the lamp's glow, insects' songs chime.[53]

This chorus of illuminated insects symbolizes vibrant vitality. Their ceaseless melodies complement the eternal illumination of the lamp, creating a scene brimming with life. Taixu finds himself rejuvenated by this dazzling spectacle.

In stark contrast to his frequent use of joyful "lamp and candle" imagery, Taixu sometimes employs similar metaphors to express negative emotions, such as anxiety about the inexorable passage of time, anguish over absent friends, and homesickness. This transition from positive to negative connotations is achieved through Taixu's skillful use of two distinct techniques: first, he reduces the number of lamps and thereby diminishes the intensity and luminosity of the light, as in such phrases as "one lamp" (*yideng* 一燈) or "lone lamp" (*gudeng* 孤燈); and, second, he places the lamp in a bleak setting. Both techniques are evident in the following passage:

獨坐寒宵盡，寒宵忽已深，一燈冷相對，惆悵去來今。

Sitting alone, the cold night wanes, In the sudden depth of the chilling hours. One lamp, cold and solitary, faces me, Regret and melancholy, the passage of time.[54]

On a frigid night, Taixu finds some solace in the company of his lamp, as if it were an old friend, silently witnessing the relentless march of time. He penned this poignant poem in 1908, at the age of nineteen, yet there is already a sense of anxiety over his advancing years. Five years later, he composed another poem that explored the same theme:

朱顏隨歲改，華髮映燈寒。

The rosy complexion changes with the passing years, The gray hair reflects the coldness of the lamp's glow.[55]

In classical Chinese literature, a person's changing complexion invariably symbolizes the unstoppable passage of time.[56] This poem was composed on the Chinese New Year's Eve of 1913—a significant moment when Taixu reflected upon his lack of purpose. The description of a pallid face and gray hair draws attention to the aging process, even though Taixu was still only twenty-three years old at the time. By exaggerating his own physical decline, he conveys a sense of loneliness and frustration in the aftermath of a turbulent period fraught with challenges and setbacks in the development of Chinese Buddhism, as well as his own life.[57]

Taixu's apprehension about the passage of time is magnified when he commemorates festive occasions, as in the following example:

等閒又度中秋節,[58]風雨孤燈思悄然！

Once again waiting idly for the Mid-Autumn Festival to arrive, With wind and rain, the solitary lamp quietly ponders.[59]

This poem was completed in 1943, near the end of Taixu's life. While others celebrate the Mid-Autumn Festival, a traditional time for family reunions, the poet sits alone, facing a single lamp, as a storm rages outside. He highlights the unrelenting passage of time, year after year, with the character *you* 又. Now an old man in a state of perpetual solitude and anxiety, does he long for some company in the lamplit night? And who is he missing most? Through his poetry, we learn that he yearns for an old friend, while the lamp bears witness to his torment:

寒窗深坐一燈昏，苦憶軒昂磊落人。

In the depths of the cold study, sitting alone in the dim lamplight, Bitter memories flood my mind of that remarkable and upright individual.[60]

The individual in question was Chen Chunbai 陳純白 (1897–1964),[61] a well-respected mayor of Hangzhou. In marked contrast to his subtle expressions of personal joy or excitement, here Taixu's yearning for his friend could scarcely be any more explicit. And he is similarly candid in a poem from 1909, evidently written during a bout of homesickness:

薄寒憶歸去，燈火見前村。

In the slight chill, memories of the journey home arise, The lamplight reveals the village ahead.[62]

As he gazes at a river, Taixu sees lamps shining in the distance, symbolizing the reunions and harmonious gatherings of countless families. This stirs up a longing for home, prompting him to declare his desire to return there. His upbringing had lacked the love and care of his parents: he had lived solely with his grandmother until leaving her behind in 1904 to become a monk, without bidding her farewell (TDQ, vol. XXXI, p. 164).[63] He had not seen her since, and the manner in which he had left his hometown had been a source of great regret, which he evokes through skillful use of "lamp and candle" imagery.

As we have seen, Taixu often interweaves "lamp and candle" imagery with other classical Chinese literary symbols—especially those associated with the night—to evoke both positive and negative emotions. Three of these groups of images are explored below.

## 5. Three Imagery Groups in Taixu's Poetry

The night provides an atmospheric backdrop against which the illumination of lamps and candles enables human activities to continue (Fu 2007, p. 251). Alongside the prominent "lamp and candle" imagery, Taixu utilizes a diverse range of imagery associated with nocturnal settings to articulate deeply held sentiments and emotions to the reader.

*5.1. Night, Fire, Boat, and Lamp (Ye Huo Chuan Deng 夜火船燈)*

There is a cohesive unity in the "night, fire, boat, and lamp" imagery group, which encapsulates multifaceted depths that resonate with the poet. The boat is a symbolic vessel, representing Taixu's lifelong journey, while the darkness of the night amplifies the intense emotions within his soul. Consequently, this imagery group bears witness to Taixu's personal wanderings, specifically capturing the essence of his melancholia during his travels:

起看繁霜白蓬背，冷冷水逼一燈寒。

[I] rise and see the boat's awning covered in frost, The chilling water presses, one lamp in solitude.[64]

In the winter of 1907, Taixu endured a restless night on a boat as the incessant cries of the crows on the riverbanks kept him awake.[65] Unable to sleep, he stepped out from beneath the boat's awning to discover a world blanketed in frost. The entire landscape shimmered as if plated in silver, yet Taixu remained untouched by its ethereal glow. A single, faint lamp illuminated his immediate surroundings, though its feeble light, reflected on the river's surface, seemed somehow diminished in the face of the cold night air. This poem vividly captures Taixu's sense of feeling lonely and adrift within the icy natural environment. Notably, he articulates these emotions through deft use of an imagery group that features in many classical Chinese poems, such as the following example:

燈影秋江寺，篷聲夜雨船。

In the temple by the autumnal river, lantern shadows flicker, Amidst the night rain, the sound of awnings echoes on the boat.[66]

This famous poem by Wen Tingyun 溫庭筠 (801–866),[67] a celebrated poet of the late Tang Dynasty, also depicts a night-time river journey. However, whereas Wen Tingyun's boat is sailing in the autumn, Taixu shifts the season to winter—a much bleaker time—and transforms the rain into frost, intensifying the harshness of the natural environment. Moreover, the gentle sound of rain on awnings is replaced by the mournful cries of crows, while the lamp, rather than illuminating the sanctuary of a temple, hangs from the boat, emphasizing Taixu's powerlessness in the face of merciless nature alongside his loneliness and vulnerability. Nevertheless, like the warm glow of a lamp piercing the darkness of a wintry night, the "night, fire, boat, and lamp" imagery group also symbolizes the poet's hope for a brighter future.

The loneliness of the traveler is a recurring theme in Taixu's work, as in the following poem that he composed during his visit to Japan in 1917:

海燈搖出西風意，漁火遠明微雪村。

The sea lantern sways, evoking the essence of the west wind, Distant fishing fires faintly illuminate the snowy village.[68]

In classical Chinese poetry, the "west wind" (*xifeng* 西風) is often synonymous with the "autumn wind" (*qiufeng* 秋風)—a metaphor for the wistfulness and yearning that are commonly associated with that season. Similarly, in Taixu's poem, the faint glow of the lantern reflects his uncertain, somewhat rudderless life. On the other hand, the distant fires of the fishermen suggest a welcoming harbor where weary travelers may find temporary respite, once again symbolizing Taixu's enduring hope. Therefore, through a poetic fusion of "night, fire, boat, and lamp" imagery, he succeeds in evoking two conflicting emotions and gives the reader a glimpse into his complex, multifaceted inner world.

*5.2. Forest, Moon, Mountain, and Lamp (Lin Yue Shan Deng 林月山燈)*

Taixu regarded the evocation of a capacious landscape as one of the pinnacles of poetic achievement.[69] By skillfully merging "lamp and candle" imagery with other classical Chinese images—such as forests, the moon, and mountains—he created an imagery group that imbues much of his poetry with a sense of vastness. Specifically, the moon symbolizes the immeasurable expanse of the sky, mountains represent the grandeur of the land,

and the flickering flames of lamps and candles within a forest signify the poet's creative endeavors. Thus, this imagery group encompasses three distinct elements—the celestial realm, the earthly domain, and the human experience—within a single, infinite space.

茂林深谷萬燈明，諰諰松濤月下鳴。

In the dense forest and deep valley, myriad lamps shine bright, The murmuring sound of pine waves[70] resonates under the moon's light.[71]

In mountain forests, lamps shed light on the limitations of human activities, creating a visual spectacle. Meanwhile, under the moonlight, resounding pine waves leave the poet awestruck. This interplay of moonlight and candlelight creates a captivating panorama of the celestial expanse above and the valley below. Once again, Taixu's debt to classical Chinese literature is apparent, as Tang poets often likened the moon to a bright lamp in order to emphasize the theme of illumination and its symbolic significance. For example:

師親指歸路，月掛一輪燈。

Under the guidance of the master, [I] am directed towards the homeward path, Where the moon hangs in the sky, resembling a radiant lamp.[72]

In this poem, Hanshan 寒山[73] narrates his journey down a mountain enveloped by swirling clouds. In the darkness of the night, he lifts his gaze and joyfully beholds the radiant moon that not only illuminates his path but also serves as a metaphorical lamp, bridging the gap between the celestial realm and the poet's earthly existence. The moon, then, is a unifying presence, harmonizing various elements to create a beautiful, complete picture. It is precisely for this reason that Taixu employs "lamp and candle" imagery to evoke thoughts of the moon in the minds of his readers:

天真爛漫聽童歌，人影燈光林外過。

With innocent delight, I listen to the children's songs, As shadows of people pass beyond the light in the woods.[74]

Taixu composed this poem in September 1925 after falling ill while delivering lectures in Singapore (TDN, p. 145).[75] During his recovery, he became intrigued by the sight of people's shadows along the edge of a forest, illuminated by the gentle glow of a lamp. Although there is no explicit mention of the moon in this poem, its radiance casts a luminous aura on Taixu's face, merging with the lamplight to illuminate his surroundings and lull him into a form of serene languor—a novel experience for the usually energetic poet. The expansive realm created by the "forest, moon, mountain, and lamp" imagery group leads the reader to the conclusion that he must have been overwhelmed by homesickness,[76] given his illness and the enforced inactivity he encountered in an unfamiliar place.[77] Hence, the poem provides a valuable insight into the subtle shifts in Taixu's personal emotions.

*5.3. Falling Leaves and Autumn Lamps (Luoye Qiudeng 落葉秋燈)*

The combination of falling leaves and autumn lamps—a prominent imagery group in both classical and modern Chinese poetry—is often used to evoke a sense of melancholy and unease. However, the free-thinking Taixu attempted to reinterpret this aesthetic cornerstone by associating it with pleasure and joy, rather than sadness and depression. The contrast is clear in the following couplets:

雨中黃葉樹，燈下白頭人。

Amidst the rain, yellow leaves adorn the trees, Like the fate of this white-haired figure beneath the lamp's glow.[78]

紅燈迴夕照，黃葉落空庭。

The red lantern casts a distant glow in the evening, While yellow leaves fall in the empty courtyard.[79]

The first couplet is taken from one of the most celebrated works in the annals of Chinese poetry.[80] It bears the imprint of Sikong Shu 司空曙 (720–790), a remarkably talented but impoverished poet of the mid-Tang era. The poem describes the events of an autumn night during which another distinguished poet, Sikong Shu's cousin Lu Lun 盧綸 (739–799),[81] pays him a visit. In the confines of Sikong Shu's humble abode, the two aged poets gaze upon raindrops falling incessantly on the windowpane as yellow leaves quiver on the trees. A dim lamp lends a pallid glow to their weathered countenances and graying hair. This forlorn scene is the setting for an elegy on the transience of time that generates an overpowering sense of despondency.[82]

In stark contrast, the second couplet, by Taixu, opens with the radiance of a red lamp, symbolizing the joy and harmony of traditional Chinese culture.[83] This crimson beacon sets a positive and uplifting tone that suffuses the whole poem, with the sunset, which typically evokes the bittersweet passage of time,[84] instilling a mood of tranquility and serenity. The falling leaves, as witnessed by Taixu, dispel the melancholy often associated with the advent of autumn, instead awakening his delight and kindling his curiosity. Therefore, although Taixu refrains from explicitly articulating any positive emotions, an unmistakable undercurrent of optimism permeates the verses.

A similar sentiment is evident in another poem he crafted during his convalescence in Singapore in 1925 (TDN, p. 145):

樹樹明燈呈幻境，窗窗涼雨寫秋痕。

The trees, each adorned with radiant lamps, present a surreal scene, While the windows, adorned with cool rain, inscribe traces of autumn.[85]

Here, Taixu rejects the traditional portrayal of a dim lamp in classical Chinese poetry and instead introduces radiant lamps that illuminate a scene of enchantment and wonder. In addition, he employs the term "*cool* rain" (*liangyu* 涼雨), rather than "*cold* rain" (*lengyu* 冷雨)—a subtle difference that suggests summer has only recently passed and autumn has not yet fully arrived—to symbolize the enduring vitality of all living things. These allusions to life and energy imbue the poem with vibrancy and optimism. In contrast, "cold rain" would have evoked later autumn—a more plaintive time of the year—as we saw earlier in his poem in remembrance of Chen Chunbai. There are echoes here of Taixu's equally nuanced descriptions of the intensity and luminosity of lamps and candles. In all instances, his aim is to convey his fluctuating emotions with absolute precision.

It is worth noting that, while Taixu's reimagining of "falling leaves and autumn lamps" imbues this imagery group with unconventional optimism, his innovation lies in reinterpreting the dynamic interplay between "lamp and candle" imagery and human emotions, rather than altering the imagery itself. Hence, meticulous analysis of his poetry is essential in order to gain full understanding of his use of this imagery to convey the peaks and troughs of his emotional landscape.

## 6. "Lamp and Candle": A Symbol of Taixu's Inner Spirit

As we have seen, Taixu employs vivid "lamp and candle" imagery in his poetry when reflecting on Buddhist teaching as well as his own heartfelt emotions. However, he also utilizes it to symbolize his inner spirit—an attribute that is best characterized as "self-sacrificing dedication" (*fen shen zhi yong* 焚身致用) (Fu 2007, p. 258). In this respect, he is a natural heir to countless classical Chinese poets who confronted the adversities and complexities of the secular world.

The essence of the imagery—a radiant light illuminating the world—harmonizes with the purpose of the mind lamp, which embodies the transmission of Buddhist wisdom. Moreover, it symbolizes Taixu's unblemished purity and unwavering integrity:

燈燈相續光無盡，塞地充天氣浩然。

The continuous glow of each lamp is boundless, Filling the earth, engulfing the vast sky.[86]

This poem was written in December 1943, after Taixu accepted an invitation from Shufang 漱芳,[87] the abbot of Huguo Temple (Huguo si 護國寺),[88] and spent a night in Hengyang 衡陽[89] (TDN, p. 329). Deeply moved by the temple's illustrious history, he composed the poem in celebration of it. The eternal radiance of each lamp symbolizes the temple's pivotal role in the evolution of Buddhism, as well as the timeless righteousness of contemporary monks, including Taixu himself. Amidst the trials and tribulations of the era, as a prominent figure amongst China's Buddhists, Taixu reflected on the darkness that shrouded the secular world.[90] Within this metaphorical shade, his virtue and honesty radiated like a guiding lamp, illuminating the path ahead. Nurturing and upholding these qualities was no simple feat, but Taixu understood the importance of persevering.

Furthermore, lamps and candles, which expend themselves to emit light, are powerful metaphors for Taixu's self-sacrifice and unwavering dedication. In this respect, his poetry perpetuates a longstanding tradition of Chinese literature that dates back two millennia, as in the following examples:

明無不見，照察纖微。以夜繼晝，烈者所依。

With brilliance that reveals all, [the lamp] illuminates even the tiniest details, Continuing day into night, it is the reliability of the resolute souls.[91]

燭之自焚以致用，亦有殺身以成仁。

The candle burns itself to serve its purpose, Sacrificing its own life to achieve benevolence.[92]

Through images of radiant lamps that illuminate the world and burning candles that fulfill their light-emitting purpose, early Chinese poets consistently highlight the importance of self-sacrifice and dedication. Taixu, a poet with a deep knowledge of the traditions of Chinese classical literature, not only embodied this spirit in his daily life but also maintained the symbolic significance of such imagery in his poetry:

千年長暗室，照破一燈寒。

In a dimly lit room, shrouded in darkness for a thousand years, One solitary lamp casts its feeble glow, piercing through the chill.[93]

Even a dim lamp may indeed illuminate a darkened chamber for as long as it burns, dedicating all its energy to lighting the room, much as Taixu commits himself to the preservation of China and the advancement of Buddhism throughout his life. This spirit of self-sacrifice finds further expression in another poem he composed on 7 July 1937—a date of great significance in China, as it was the day on which the Lugouqiao Incident (*Lugouqiao shibian* 盧溝橋事變) took place:[94]

心海騰宿浪，風雨逼孤燈。

The vast sea of the heart surges with restless waves, As winds and rains threaten the solitary lamp.[95]

When Taixu heard of the Lugouqiao Incident, he was enveloped by profound and overwhelming sorrow (*bei kai wusi* 悲慨無似) (TDN, p. 271). As a deeply conscientious monk, he remained steadfast in his commitment to both the welfare of his homeland and the promotion of Buddhism.[96] However, the incident heralded a period of immense suffering for the Chinese people, coupled with a hiatus in the development of Buddhism. Both were reflected in Taixu's subsequent poetry, as he started to question Buddhism's capacity to rescue the nation and secure its own future.[97] Nevertheless, he continued to work selflessly for both causes, undeterred by increasing physical and mental frailty.[98] In the face of this adversity, he more than ever resembled a solitary lamp glowing dimly but resolutely amidst biting wind and persistent rain.

It is this inherent dimness that imbues the "lamp and candle" imagery in Taixu's poetry with such a resonant theme of tragedy. As mentioned earlier, he often employs such imagery—and especially depictions of a single lamp—to evoke negative emotions, such as homesickness, loneliness, and anxiety in relation to the passage of time. Indeed, even his

prose contains a number of poignant narratives that revolve around this imagery, as in the following example:

> 譬如然燈，膏油既盡，不久將滅。老亦如是，壯膏既盡，不久將死。

> Metaphorically speaking, just like a lamp, once its oil is exhausted, will soon extinguish, so is the nature of aging. When one's vitality and vigor are depleted, death will not be far behind.[99]

This is Taixu's interpretation of the concept of *laoku* 老苦 (*jarā-duḥkha*),[100] which symbolizes the hardships of old age. With the passage of time, the elderly poet Taixu, like a lamp burning its final drops of oil, moves inexorably toward the realm of death. Moreover, his frequent use of the phrase "wind candle" (*fengzhu* 風燭) in his later works conveys a deep understanding of the inherent tragedy of his own destiny:

> 風燭無常願無盡，海天雲水正茫茫。

> The fleeting nature of the wind candle embodies boundless aspirations, The vastness of the sea, sky, clouds, and water appears infinite and profound.[101]

Taixu composed this poem in Berlin on his fortieth birthday in 1928. In it, he strikingly juxtaposes the vastness of the natural world with the humble, flickering candle that symbolizes his own isolation, vulnerability, and helplessness. Despite nurturing numerous aspirations, he must confront the reality of his own advancing years and make grave decisions regarding how best to dedicate himself to the causes of China and Buddhism.

The same image again evokes the fleeting nature of human existence in one of Taixu's final texts, an elegy for his cherished disciple Fushan 福善 (1915–1947),[102] which he composed in 1947:

> 現在，風燭殘命的我，仍風中燭似的殘存著。

> In the present moment, I, like a wind-blown candle, continue to exist with a flickering flame.[103]

Fushan had declared his intention to care for the aging Taixu (TDQ, vol. XXXIII, p. 226), but it was the younger man who died first—from smallpox on 20 February 1947. The profound grief that Taixu experienced due to the loss of his beloved student compounded the sorrow he felt for the fate of Chinese Buddhism (TDN, p. 348). Less than a month later, on March 17, he suffered stroke while delivering a lecture and his remarkable life came to an end (TDN, p. 349).

There is undoubtedly a hint of regret in some of Taixu's reflections on his life of self-sacrifice, as is evident in a poem in which he dedicates himself unconditionally to the Buddha:

> 仰止唯佛陀，完就在人格。

> In profound meditation, the sole pursuit is the Buddha, Complete fulfillment resides within the virtuous nature of humanity.[104]

Nevertheless, his final poem, *Feng Zangweng* 奉奘翁 (*Poem Dedicated to Master Zang*),[105] demonstrates that he never abandoned his dual, self-imposed mission to safeguard the nation's well-being and advance Buddhism. He remained a deeply emotional and selfless monk and poet until the very end of his life. Indeed, these are the qualities that his readers have always most admired in him.

## 7. Conclusions

Taixu not only inherited but also creatively developed classical Chinese "lamp and candle" imagery in his poetry. This artistic choice was deeply rooted in his personal affinity for the imagery and nurtured by his upbringing in an environment steeped in classical Chinese literature, and especially Tang poetry, in which these sources of light carry enormous symbolic significance. As a devoted monk, Taixu skillfully employs this imagery as a metaphor for both Buddhist wisdom and transmitting the Buddhist Dharma. Simultaneously, as a skilled poet, he adeptly communicates his personal sentiments—encompassing

the full panoply of positive and negative human emotions—by drawing on a diverse array of classical Chinese imagery, from lamps and candles to boats, the moon, and falling leaves.

The recurring themes of burning and illumination, which lie at the heart of this imagery, provide invaluable insights into Taixu's defining characteristics—purity, integrity, devotion, and self-sacrifice. Yet, they suggest that he viewed himself not only as a virtuous Chinese Buddhist leader but also as a gifted poet. This aspect of his personality certainly warrants further scholarly exploration.

In conclusion, Taixu's poetic engagement with "lamp and candle" imagery exemplifies his mastery of classical Chinese poetic tropes while also offering thought-provoking insights into the human condition. It serves as a conduit for transmitting Buddhist wisdom and simultaneously affords a glimpse into his complex inner world. Through his deft use of this imagery, Taixu exhibits a unique fusion of spiritual devotion, artistic sensibility, and emotional depth. It is to be hoped that further study will help to unravel the intricacies of his poetic legacy and clarify its enduring impact on Chinese literature and modern Buddhism.

**Funding:** This research was funded by China Scholarship Council (No. 202006780014). And The APC was funded by the Special Research Fund (BOF) of Ghent University (No. 01SC0120).

**Institutional Review Board Statement:** Not applicable.

**Informed Consent Statement:** Not applicable.

**Data Availability Statement:** Not applicable.

**Conflicts of Interest:** The author declares no conflict of interest.

### Abbreviations

TDQ    (Shi 2005)
TDN    (Shi 2011)

## Notes

[1] The Buddhist reform movement was a collaborative effort involving various groups, including reformists such as Kang Youwei 康有為 (1859–1927), Liang Qichao 梁啟超 (1873–1929), Tan Sitong 譚嗣同 (1865–1898), and Zhang Taiyan 章太炎 (1869–1936), lay Buddhists like Yang Renshan 楊仁山 (1837–1911) and Ouyang Jian 歐陽漸 (1871–1943), and Buddhist monks such as Jing'an 敬安 (1851–1912), Taixu 太虛, Xuyun 虛雲 (1840–1959), and Su Manshu 蘇曼殊 (1884–1918). For more information related to Chinese modern Buddhist reform movements, refer to the following three books: Tarocco (2005); Jessup and Kiely (2016); Campo and Bianchi (2023).

[2] According to Taixu: "In the realm of Chinese poetry, the Tang Dynasty reigns supreme" 中國詩以唐為盛 (TDQ, vol. XXXII, p. 414).

[3] The TDQ contains approximately 500 poems by Taixu's mentors, companions, disciples, and students. All of these works are intimately connected with Taixu himself, encompassing verses presented to him as well as collaborative compositions. For further details, (see TDQ, vol. XXXIV, pp. 290–444).

[4] *Meian shilu* soon gained many readers. Their support is reflected in the prefaces of the TDQ, which were composed by several of Taixu's fellow poets and friends (TDQ, vol. XXXII, p. 510).

[5] Yinshun was a renowned Buddhist philosopher who joined Taixu in the modern Buddhist revival movement in 1930. Throughout the rest of his life, he dedicated himself to promoting "humanistic Buddhism" (*renjian fojiao* 人間佛教), which encompassed many of the concepts and principles advocated by Taixu. For more in-depth study of Yinshun (see Bingenheimer 2009; Lee 2021).

[6] There are several important works dedicated to the study of Taixu. Welch (1968) devotes a chapter to Taixu, presenting him as a disingenuous self-promoter. Jiang (1993) provides a balanced perspective on the first half of Taixu's life. Hong (1999) takes a thematic approach to examining Taixu's activities and contributions. Pittman (2001) is considered a significant work, delving into Taixu's efforts to make Chinese *Mahāyāna* Buddhism relevant to the modern world. Goodell (2008) sheds light on Taixu's seminal period of life and thoughts. Ritzinger (2017) focuses on Taixu's Buddhist radicalism. Jones (2021) regards Taixu as a transitional figure in the establishment of a "Pure Land in the Human Realm" (*renjian jingtu* 人間淨土).

[7] For the study of "lamp and candle" imagery in classical Chinese poetry, there are two notable works. Tian (2005) offers a Buddhist perspective to study Liang (502–557) poetry through the lens of this imagery. Fu (2007), in the chapter titled "Zhuguang dengying

li de zhongguo shi" 燭光燈影裏的中國詩 (Chinese Poetry in the Glow of Candlelight and Lamplight), pp. 231–61, provides a comprehensive examination of the literary and religious significance of this imagery in Tang poetry.

8     In modern China, many monks have shown a preference for incorporating "lamp and candle" imagery into their poetry. For instance, Xuyun, in his poem *Zai Jilongpo Lingshan si Yang Shaohong laifang buyu*在吉隆玻靈山寺楊少洪來訪不遇 (*Yang Shaohong's Unmet Visit to Lingshan Temple in Kuala Lumpur*), poetically employs the lamp as a symbol of hope for a new day: "Under the lamp, I read repeatedly through the night/Unknowingly, the eastern window gradually reveals a tinge of red" 夜來燈下頻頻讀，不覺東窗漸透紅. Jing'an, in his poem *Zixiao shi* 自笑詩 (*Self-Mockery of Poetry*), utilizes the imagery to depict his self-sacrifice for Buddhism: "Sacrificing flesh, lighting the lamp for the Buddha's service / Realizing that the body is but a bubble in water" 割肉燃燈供佛勞，了知身是水中泡. Also Hongyi 弘一 (1880–1942), in his poem *Xijiang yue Su Tanggu lüguan* 西江月 宿塘沽旅館 (*The Moon over West River*), employs the lamp to convey his sense of solitude in the night: "The remaining trickle startles within a dream/A solitary lamp and the scenery form a pair" 殘漏驚人夢裏，孤燈對景成雙. Su Manshu, in his poem *Dongju* 東居 (*Eastern Abode*), sensitively captures the ambiance through descriptions of the lamp in autumn: "The lamp floats amidst beaded curtains, the jade zither resonates in the autumn/Several melodies echo at the waterside pavilion" 燈飄珠箔玉箏秋，幾曲回闌水上樓.

9     According to Taixu, the Dayin Nunnery was situated approximately three miles from Chang'an 長安, Taixu's birthplace (TDQ, vol. XXXI, p. 155).

10    Taixu, "Shengzhang Zai Nong Gong Dao Shang Du De Xiangzhen" 生長在農工到商讀的鄉鎮 (Growing up in a Town: From Farming and Laboring to Commerce and Education), in *Taixu zizhuan* 太虛自傳 (*Taixu's Autobiography*) (TDQ, vol. XXXI, p. 156).

11    The Sacred Heart Basilica (Basilique du Sacré-Cœur) is one of Paris's most prominent landmarks.

12    In 1925, Taixu decided to travel to Europe as a Buddhist missionary. He and his companions began their journey on 11 August 1928, and arrived in Paris on September 16. He remained in the city for more than a month, during which time he delivered lectures to various political and academic associations (See TDQ, vol. XXXI, pp. 326–27, 334–42).

13    Taixu. *Bali jiyou* 巴黎紀遊 (*Travelogue of Paris*) (TDQ, vol. XXXIV, pp. 135–37).

14    The Forbidden City served as the imperial residence for twenty-four emperors of the Ming and Qing dynasties, and it is considered to be the largest and most complete architectural complex of its kind still in existence.

15    The Putuo Mountain, in the Zhoushan 舟山 Archipelago, Zhejiang Province 浙江省, is a renowned center of Buddhist pilgrimage and worship.

16    Taixu. *You Shanghai di Mensi manyin* 由上海抵門司漫吟 (*A Rambling Poem from Shanghai to Moji Ward*) (TDQ, vol. XXXIV, p. 75). Moji is a ward of the city of Kitakyushu in Fukuoka Prefecture, Japan.

17    Yingjiang Temple in Anqing, Anhui Province 安徽省, was established in 1619. The Zhenfeng Pagoda, a seven-story tower within the temple, was previously known as the "Tower of Ten Thousand Buddhas" (Wanfo ta 萬佛塔).

18    Taixu. *Yingjiang si Zhenfeng ta* 迎江寺振風塔 (*The Zhenfeng Pagoda of Yingjiang Temple*) (TDQ, vol. XXXIV, p. 103).

19    A *mengguan* ("hall for untaught children") was essentially a primary school that taught boys how to read and write in preparation for imperial examinations (See Brokaw 2020).

20    The *Quan Tangshi* anthology was completed in 1706. It includes 49,403 poems written by 2873 poets during the Tang Dynasty.

21    The poems of Bai Juyi (also known by his courtesy name Letian 樂天) are renowned for their accessibility and clarity.

22    Bai Juyi. *Xizeng lijing laoseng* 戲贈禮經老僧 (*A Playful Tribute to the Elderly Monk who Meditates and Recites Scripture*) (Xie 2006, p. 2646).

23    In Buddhist philosophy, the mind lamp concept symbolizes the attainment of mental clarity and illumination that arises from a state of stillness and tranquility (See Ding 2012, p. 710).

24    The association in question was the Buddhist Alliance for Preservation (Weichi Fojiao Tongmenghui 維持佛教同盟會), which Taixu hoped to establish in March 1913. However, he abandoned the idea following discussions with friends, who perceived it as no different from any number of existing organizations (See TDN, p. 39).

25    Taixu. *Weichi Fojiao Tongmenghui xuanyan* 維持佛教同盟會宣言 (*Declaration of the Buddhist Alliance for Preservation*) (TDQ, vol. XXXIII, p. 6).

26    In Taixu's perspective, the sun, the moon, and the lamp serve as three illuminating symbols, reflecting the profound wisdom and luminosity of the Buddha's *tathāgatagarbha* (*rulai zangxin* 如來藏心). These illuminants are metaphorically associated with the Buddha's radiant fruition: the sun symbolizes the merit of the Buddha's wisdom (*zhide* 智德), the moon represents the merit of severing afflictions (*duande* 斷德), and the lamp embodies the merit of the Buddha's compassion (*ende* 恩德) (See TDQ, vol. XI, p. 78).

27    Taixu. *Fahua jiangyan lu* 法華講演錄 (*Recorded Lectures on the Lotus Sūtra*) (TDQ, vol. XI, p. 78).

28    Taixu. *Yaoshi liuli guang rulai benyuan gongde jing jiangji* 藥師琉璃光如來本願功德經講記 (*Annotated Lectures on the Original Vows of the Medicine-Master Tathāgata of Lapis Light*) (TDQ, vol. XV, p. 387).

29    Taixu. *He Zhan'an guo Hanyang Guiyun Si diaoyun yan*和湛庵過漢陽歸元寺吊雲岩 (*A Poetic Tribute to Zhan'an at Guiyuan Temple in Hanyang, When Observing the Diaoyun Yan*) (TDQ, vol. XXXIV, p. 63).

30 Huoxuan, also known as Zhan'an 湛庵, is a poet-monk who shares a profound friendship with Taixu, forged through their mutual appreciation of each other's poetry (TDQ, vol. XXXI, p. 184). His poetic exchanges with Taixu are preserved in the latter's poetry collections and a short biography (TDQ, vol. XXXIII, pp. 299–302).

31 Lingyun Temple, in Taizhou 臺州, Zhejiang Province, is located in a breathtaking natural landscape, surrounded by picturesque mountains.

32 Taixu. *Wansu Taohua Lingyun si* 晚宿桃花靈雲寺 (*An Evening Stay at Lingyun Temple, Where Peach Blossoms Flourish*) (TDQ, vol. XXXIV, p. 246).

33 Taixu. *Wo zhi zongjiao guan* 我之宗教觀 (*My Religious Views*) (TDQ, vol. XXII, p. 225).

34 Taixu. *Sanbao ge* 三寶歌 (*Song of the Three Treasures*) (TDQ, vol. XXXIV, p. 262).

35 According to Taixu, Shi Ji'an was an exceptionally talented poet who deserved the title "poetic master" (*shi zhi cizong* 詩之詞宗) (See TDQ, vol. XXXI, p. 305).

36 On 11 November 1917, following the conclusion of the Buddhist assembly in Zhanghua 彰化, Taixu joined local officials and journalists at a poetry gathering (TDN, p. 59). The term 文字 refers to the poems the participants composed during that meeting.

37 "Conditioned phenomena" (*youwei fa* 有為法) are manifestations of causes and conditions.

38 Taixu. *Zhanghua Tanhua tang jixi da Shi Ji'an* 彰化曇華堂即席答施寄庵 (*Spontaneous Response to Shi Ji'an's Poem at the Tanhua Hall in Zhanghua*) (TDQ, vol. XXXIV, p. 82). This poem is a response to Shi Ji'an's *Zeng Taixu fashi* 贈太虛法師 (*A Poem Dedicated to the Venerable Taixu*), ibid., p. 311.

39 Taixu. *Jiuhua zashi shishou* 九華雜詩十首 (*Ten Miscellaneous Poems on Jiuhua*) (TDQ, vol. XXXIV, p. 143).

40 Taixu wrote numerous articles on the subject of Chinese Buddhist education, including: *Zhongguo de seng jiaoyu ying zenyang* 中國的僧教育應怎樣 (*How Should Buddhist Education in China Be?*) (TDQ, vol. XIX, p. 31); *Xiandai xuyao de seng jiaoyu* 現代需要的僧教育 (*Contemporary Needs of Buddhist Education*), ibid., p. 37; and *Fojiao yingban zhi jiaoyu yu seng jiaoyu* 佛教應辦之教育與僧教育 (*Education in Buddhism and Buddhist Education that Should Be Established*), ibid., p. 20. For more information about Taixu's modern Buddhist education (see Li 2013; Travagnin 2017; Lai 2017).

41 The *Ben Cao Gang Mu*, compiled by the distinguished medical scientist, pharmacist, and naturalist Li Shizhen 李時珍 (1518–1593), is a comprehensive collection of Chinese materia medica from ancient times to the sixteenth century.

42 Li Shizhen, "*zhujin* 燭燼, candle remains" (Li 2021, p. 135).

43 *Yi Wen Zhi*, a significant bibliographic work compiled by Ban Gu 班固 (32–92) that forms part of the *Hanshu* 漢書 (*History of the Former Han Dynasty*), draws heavily upon Liu Xin's 劉歆 (50 BCE–23 CE) *Qishu* 七書 (*Seven Summaries*).

44 Taixu. *You Taibei jie* 遊臺北街 (*Exploring the Streets of Taipei*) (TDQ, vol. XXXIV, p. 80).

45 In September 1917, Shanhui 善慧 (1881–1945), the abbot of Lingquan Monastery (Lingquan si 靈泉寺) on Yuemei Mountain 月眉山, Jilong 基隆, Taiwan, expressed his intention to organize a Buddhist assembly. He invited Yuanying 圓瑛 (1878–1953) to deliver a lecture, but the latter was unable to attend and suggested Taixu should take his place. Taixu had already planned his trip to Japan, so he decided to break his journey in Taiwan in order to attend the assembly (See TDN, p. 58).

46 Kyoto holds considerable historical and cultural significance as a political center of Japan from the Middle Ages to modern times. It served as the country's capital from 794 to 1869.

47 Taixu. *Jingdu you* 京都遊 (*A Visit to Kyoto*) (TDQ, vol. XXXIV, p. 91).

48 Taixu. *Jiuhua zashi shishou* 九華雜詩十首 (*Ten Miscellaneous Poems on Jiuhua*) (TDQ, vol. XXXIV, p. 143).

49 Taixu. *Guan Zongxian ju* 觀宗閒居 (*Observing the Leisure Dwelling*) (TDQ, vol. XXXIV, p. 236).

50 Taixu. *Xihu xiaye* 西湖夏夜 (*A Summer Night at West Lake*) (TDQ, vol. XXXIV, p. 129).

51 West Lake is on the western side of old Hangzhou 杭州. Bai Juyi is credited with coining the appellation in one of his poems.

52 In ancient Chinese literature, *jinsu* 金粟 typically denotes the wick of a lamp. In the context of this poem, it symbolically represents the Buddha.

53 Taixu. *Jiuhua zashi shishou* 九華雜詩十首 (*Ten Miscellaneous Poems on Jiuhua*) (TDQ, vol. XXXIV, p. 143).

54 Taixu. *Shenye* 深夜 (*Late Night*) (TDQ, vol. XXXIV, p. 11).

55 Taixu. *Guichou chuxi* 癸醜除夕 (*New Year's Eve of the Year Gui Chou*) (TDQ, vol. XXXIV, p. 48).

56 In *Die lian hua: Yuejin tianya libie ku* 蝶戀花·閱盡天涯離別苦 (*Butterflies in Love with Flowers: Experiencing the Bitterness of Farewell across the Ends of the Earth*), a *ci* 詞 poem by Wang Guowei 王國維 (1877–1927), one verse expresses the transience of life: "The passage of time cannot be detained / Rosy cheeks bid farewell to the mirror, flowers to the tree" 最是人間留不住，朱顏辭鏡花辭樹. Wang Guowei, a renowned scholar and poet, hailed from Taixu's hometown—Haining 海寧, Zhejiang Province.

57 During the early Republican period, Taixu underwent a series of tumultuous experiences that greatly influenced his life and work. (See TDQ, vol. XXXI, pp. 188–92).

58 The Mid-Autumn Festival (*Zhongqiu jie* 中秋節), observed on the fifteenth day of the eighth lunar month, holds great significance in Chinese culture. It celebrates the reunion of loved ones, often represented by the full moon, and is infused with a profound sense of longing for one's hometown and relatives.

59 Taixu. *Renwu Zhongqiu Guanyue ting*壬午中秋觀月亭 (*Observing the Moon Pavilion on the Mid-Autumn Festival in the Year of Renwu*) (TDQ, vol. XXXIV, p. 254).

60 Taixu. *Huai Chen Chunbai* 懷陳純白 (*In Remembrance of Chen Chunbai*) (TDQ, vol. XXXIV, p. 48).

61 Chen Chunbai was a native of Yongjia 永嘉, Zhejiang Province.

62 Taixu. *Qiujiang wantiao* 秋江晚眺 (*Autumn Evening Gazing by the Riverside*) (TDQ, vol. XXXIV, p. 13).

63 Taixu's cherished grandmother exerted a profound and enduring influence on his life and literary career (See TDQ, vol. XXXI, pp. 155–56, 161–62).

64 Taixu. *Zhouzhong muye* 舟中莫夜 (*Embracing the Night in the Boat*) (TDQ, vol. XXXIV, pp. 9–10).

65 In the same poem, Taixu writes: "The remaining drops of the night's hourglass drip, and the dreams wane in the haze/The crows' cries on both riverbanks penetrate the ears with a poignant sorrow" 滴殘更漏夢闌珊，兩岸烏啼入耳酸 (See TDQ, vol. XXXIV, p. 9).

66 Wen Tingyun. *Songseng dongyou* 送僧東遊 (*Sending the Monk on an Eastern Journey*) ([Liu 2016](), p. 427).

67 Wen Tingyun was a native of Taiyuan 太原 in Shanxi Province 山西省. His elegant, rhythmical, exquisitely crafted poetry often focuses on the themes of sorrow and loss.

68 Taixu. *Bo Mensi* 泊門司 (*Anchored at Moji Ward*) (TDQ, vol. XXXIV, p. 93).

69 Many of Taixu's predecessors shared this opinion, especially during the high point of Tang poetry between the first year of the Kaiyuan 開元 era and the fourteenth year of the Tianbao 天寶 era (713–755). For instance, an esteemed poet of that period, Wang Wei 王維 (692–761), captured the imagination of both contemporary and future generations through his depiction of a seemingly infinite wasteland in *Shi zhi saishang* 使至塞上 (*On a Mission to the Frontier*). In particular, the famous lines "In a boundless desert lonely smoke rises straight/Over an endless river the sun sinks round 大漠孤煙直，長河落日圓" vividly evoke the grandeur and splendor of this vast landscape, leaving an indelible impression on the reader. Three centuries later, a scholar–official of the Northern Song Dynasty, Fan Zhongyan 范仲淹 (989–1052), revisited this theme in his *Yujiaao Qiusi* 漁家傲秋思 (*The Pride of Fishermen*): "All hills low/Dust touches the town with hue" 千嶂裏，長煙落日孤城閉. A third example is provided by Li Panlong 李攀龍 (1514–1570), a Ming Dynasty literatus, in his poem *Guangyang shan daozhong* 廣陽山道中 (*The Path of Guangyang Mountain*): "The thunderous roar descends upon a thousand peaks/The rain-hued colors arrive on myriad mountains" 雷聲千嶂落，雨色萬峰來.

70 A "pine wave" is the rhythmical sound of wind sweeping through a pine forest.

71 Taixu. *Yuexia you Guling fan Zhibi feng hui Dalin*月下由牯嶺翻擲筆峰回大林 (*Moonlight from Guling over the Fanzhibi Peak to Dalin*) (TDQ, vol. XXXIV, p. 115).

72 Hanshan. *Shi Sanbai sanshou qi yiliuliu* 詩三百三首其一六六 (*Three Hundred and Three Poems: Poem One Hundred and Sixty-Six*) ([Xiang 2000](), p. 434). In the annotation of this poem, Xiang Chu項楚 provides a detailed explanation regarding the historical context and rationale behind the comparison of the moon to a lamp.

73 Hanshan lived in seclusion at Guoqing Monastery (Guoqing si 國清寺) on Tiantai Mountain (Tiantai shan 天臺山). He practiced a unique amalgamation of Confucianism, Taoism, and Buddhism, reflecting a harmonious synthesis of all three influential schools of thought.

74 Taixu. *Oucheng* 偶成 (*Occasional Composition*) (TDQ, vol. XXXIV, p. 249).

75 Taixu traveled to Singapore in July 1925 in preparation for the much more ambitious journey to Europe and America that he would undertake three years later (See TDQ, vol. XXXI, p. 273).

76 In classical Chinese literature, the moon, like the lamp, is often used to convey homesickness. For instance, in his poem *Jingye si* 靜夜思 (*A Tranquil Night*), Li Bai 李白 (701–762) writes: "Looking up, I find the moon bright/Bowing, in homesickness I drown" 舉頭望明月，低頭思故鄉.

77 This notion is supported by the fact that Taixu resolved to return to China after a mere two weeks of recovery (See TDN, p. 147).

78 Sikong Shu. *Xi waidi Lu Lun jiansu* 喜外弟盧綸見宿 (*Glad to See Cousin Lu Lun's Overnight Stay*) ([Wen 2011](), p. 286).

79 Taixu. *Xianju* 閒居 (*Inactivity*) (TDQ, vol. XXXIV, p. 128).

80 In *Siming shi hua* 四溟詩話 (*Siming Poetry-Talk*), a treatise on poetry dating from the Ming Dynasty, Xie Zhen 謝榛 (1495–1575) cites Sikong Shu's poem as the finest example of a poet's use of the falling leaves and autumn lights imagery group, placing it above similar works by Wei Yingwu 韋應物 (731–791) and Bai Juyi in terms of its powerful depiction of the season.

81 Lu Lun, a poet of the Tang Dynasty, faced a lack of success in his literary career. His poetic compositions primarily revolved around the themes of presentation and response, while also offering insights into the realities of life within the army.

82 Sikong Shu's verses are frequently melancholic, especially when he reflects on the aftermath of war.

83 In classical China poetry, the candle is often used in place of a red lamp when depicting joyous or festive occasions. A notable example may be found in Li Shangyin's 李商隱 (813–858) *Huaxia zui* 花下醉 (*Intoxicated with Flowers*), in which he gazes upon flowers after an evening drinking with friends: "Guests departed, now sober amidst the late hours/Holding a candle, [I] venture to behold the excessive blossoms" 客散酒醒深夜後，更持紅燭賞殘花.

84    There is a fine example in Li Shangyin's *Deng Leyou yuan* 登樂遊原 (*Atop Mount Leyou*): "The setting sun seems so sublime/Yet it nears its waning hours" 夕陽無限好，只是近黃昏.

85    Taixu. *Daguan yuan jijing* 大觀園即景 (*Observations of the Grand View Garden*) (TDQ, vol. XXXIV, p. 249).

86    Taixu. *Shufang yaosu Hengyang Huguo si* 漱芳邀宿衡陽護國寺 (*Invitation to Reside at Hengyang Huguo Temple by Shufang*) (TDQ, vol. XXXIV, p. 232).

87    Shufang, who had been one of Taixu's students, played a significant role in the latter's activities in Hengyang. In December 1943, when Taixu became abbot of Huayao Temple (Huayao si 花藥寺), he undertook the reorganization of the Hengyang Buddhist Association (Hengyang fojiao hui 衡陽佛教会) and appointed Shufang as its president.

88    Huoguo Temple, also known as Jiulong Monastery (*Jiulong an* 九龍庵), was originally constructed in 1579 and underwent several reconstructions over subsequent centuries. However, it was destroyed during the Cultural Revolution (*Wenhua dageming* 文化大革命; 1966–1976).

89    Hengyang is in the south of Hunan Province 湖南省. On 6 December 1943, Taixu visited the city in order to promote the Buddhist Dharma. He received a warm welcome from his students, including Shufang (TDN, pp. 329–30).

90    China was in the midst of the most challenging phase of the Second Sino-Japanese War (1931–1945) at this time.

91    Liu xin 刘歆 (50 BCE–23 CE). *Deng Fu* 灯赋 (*Ode to the Lamp*). (Yan 1999a, p. 410).

92    Fu Xian 傅鹹 (239–294). *Zhu Fu* 燭賦 (*Ode to the Candle*) (Yan 1999b, pp 533).

93    Taixu. *Zeng Huang Baoguang* 贈黃葆光 (*A Poem to Huang Baoguang*) (TDQ, vol. XXXIV, p. 126).

94    The Lugouqiao Incident, a military confrontation between Chinese and Japanese troops at Lugouqiao 盧溝橋 in Wanping County 宛平縣, Hebei Province 河北省, is widely regarded as the catalyst that ignited the Second Sino-Japanese War—a protracted and devastating conflict between the two nations.

95    Taixu. *Lushan zhu mao jishi* 廬山住茆即事 (*Observations of Residing in a Bothy on Mount Lu*) (TDQ, vol. XXXIV, p. 190). Taixu was living on Mount Lu (Lushan 廬山), a prominent Buddhist mountain in Jiangxi Province 江西省, at the time of the Lugouqiao Incident.

96    In *Lushan zhu mao jishi*, Taixu writes: "For three decades, I have borne the worries of the world/For twenty years, I have dedicated myself to the salvation of monks" 卅載知憂世，廿年勵救僧.

97    Also in *Lushan zhu mao jishi*, Taixu writes: "In the end, witnessing the bravery of demons/Withheld is the declaration that the Buddha lacks power" 終看魔有勇，忍說佛無能.

98    In *Sanbao ge* 三寶歌 (*Song of the Three Treasures*), composed toward the end of his life, Taixu writes: "With fullness of life, dedicating [my] being and destiny/In faith and acceptance, diligently fulfilling [my] duty" 盡形壽，獻身命，信受勤奉行 (See TDQ, vol. XXXIV, p. 262).

99    Taixu. *Dacheng liqu liu boluomiduo jing guiyi sanbao pin jianglu* 大乘理趣六波羅蜜多經皈依三寶品講錄 (*A Discourse on the Chapter "Taking Refuge in the Three Treasures" in the Dasheng liqu liu boluomiduo jing*) (TDQ, vol. IV, p. 44).

100   *Laoku* is one of the four sufferings (*siku* 四苦) associated with birth, old age, sickness, and death.

101   Taixu. *Sishi chudu zai Bolin sheying ziti* 四十初度在柏林攝影自題 (*Self-Portrait Taken in Berlin at Forty*) (TDQ, vol. XXXIV, p. 139).

102   Fushan, who displayed a remarkable intellect from a very young age, entered the monastic life at the age of thirteen and became one of Taixu's students in August 1937.

103   Taixu. *Tong Fushan* 慟福善 (*Sorrowful Death of Fushan*) (TDQ, vol. XXXIII, p. 226).

104   Taixu. *Man sishi ba shuoji huixiang wai zumu Zhang-Zhou shi mu Lüshi huozeng anle*滿四十八說偈回向外祖母張周氏母呂張氏獲增安樂 (*A Discourse in Verse on Turning Forty-Eight, Offering Dedication to Grandmother Zhang-Zhou and Mother Zhang for Increased Peace and Happiness*) (TDQ, vol. XXXIV, p. 262).

105   In February 1947, Taixu returned to Ningbo to visit one of his masters, Zangnian 奘年. It was after this meeting—which Yinshun pointedly terms a "farewell" (*you juebie zhi zhao* 有訣別之兆)—that he composed *Feng Zang Weng*, including the lines "With unwavering dedication and simplicity, a lifetime promoting Buddhism/Transcending worldly affairs, freely and effortlessly" 勤樸一生禪誦力，脫然瀟灑出凡塵. There is a sense of urgency, of time slipping away, in these lines, coupled with a hint of sadness as Taixu reflects on the future of Chinese Buddhism after his master's—and possibly his own—death. Moreover, he draws an implicit contrast between Zangnian's selfless magnanimity and his own failure to find solace at the end of a life of devoted service. This echoes the sentiments of some of his earlier writings, where he occasionally describes his Buddhist endeavors as a "loss" (See TDN, p. 346).

## Archival Sources

1913. *Weichi Fojiao Tongmenghui xuanyan* 維持佛教同盟會宣言 (*Declaration of the Buddhist Alliance for Preservation*). TDQ: XXXIII: 3–6.

1917. *Dongying caizhen lu* 東瀛采真錄 (*Record of Gathering Truths in the Eastern Land*). TDQ: XXXI: 287–325.

1921. *Fahua jiangyan lu* 法華講演錄 (*Recorded Lectures on the Lotus Sutra*). TDQ: XI: 77–80.

1925. *Wo de zongjiao guan* 我的宗教觀 (*My Religious Views*). TDQ: XXII: 219–227.

1929. *Huanyou ji* 寰遊記 (*A Travelogue Around the World*). TDQ: XXXI: 326–396.

1933. *Dacheng Liqu Liu Boluomiduo Jing Guiyi Sanbao Pin Jianglu* 大乘理趣六波羅蜜多經皈依三寶品講錄 (*A Discourse on the Chapter "Taking Refuge in the Three Treasures" in the* Dasheng Liqu Liu Boluomiduo Jing). TDQ: IV: 3–55.

1934. *Yaoshi liuli guang Rulai benyuan gongde jing jiangji* 藥師琉璃光如來本願功德經講記 (*Annotated Lectures on the Original Vows of the Medicine-Master Tathāgata of Lapis Light*). TDQ: XV: 385–387.

1938. *Fojiao yingban zhi jiaoyu yu seng jiaoyu* 佛教應辦之教育與僧教育 (*Education in Buddhism and Buddhist Education that Should Be Established*). TDQ: XIX: 20–27.

1938. *Seng jiaoyu zhi mudi yu chengxu* 僧教育之目的與程式 (*The Purpose and Process of Monastic Education*). TDQ: XIX: 14–19.

1938. *Xiandai xuyao de seng jiaoyu* 現代需要的僧教育 (*Contemporary Needs of Buddhist Education*). TDQ: XIX: 37–40.

1938. *Zhongguo de seng jiaoyu ying zenyang* 中國的僧教育應怎樣 (*How Should Buddhist Education in China Be*). TDQ: XIX: 31–36.

1939. *Taixu zizhuan* 太虛自傳 (*Taixu's Autobiography*). TDQ: XXXI: 151–284.

1946. *Shicun* 詩存 (*Poetry Collection*). TDQ: XXXIV: 3–289.

1947. *Tong Fushan* 慟福善 (*Sorrowful Death of Fushan*). TDQ: XXXIII: 223–226.

## Published Sources

Bingenheimer, Marcus. 2009. Writing History of Buddhist Thought in the Twentieth Century: Yinshun (1906–2005) in the Context of Chinese Buddhist Historiography. *Journal of Global Buddhism* 10: 255–90.

Brokaw, Cynthia J. 2020. *Commerce in Culture: The Sibao Book Trade in the Qing and Republican Periods*. Cambridge: Harvard University Press.

Campo, Daniela, and Ester Bianchi. 2023. *"Take the Vinaya as Your Master". Monastic Discipline and Practices in Modern Chinese Buddhism*. Amsterdam: Brill.

Ding, Fubao 丁福保. 2012. *Foxue da cidian* 佛學大辭典 (*The Comprehensive Dictionary of Buddhist Studies*). Taipei: Hwadzan Pure Land Association 華藏淨宗學會.

Fu, Daobin 傅道彬. 2007. *Wan Tang zhongsheng: Zhongguo wenxue de yuanxing piping* 晚唐鐘聲：中國文學的原型批評 (*The Bell Sound of Late Tang: A Prototypical Critique of Chinese Literature*). Beijing: Peking University Press 北京大學出版社.

Goodell, Eric. 2008. Taixu's Youth and Years of Romantic Idealism, 1890–1914. *Chung-Hwa Buddhist Journal* 21: 77–121.

Hong, Jinlian 洪金蓮. 1999. *Taixu dashi: Fojiao xiandaihua zhi yanjiu* 太虛大師佛教現代化之研究 (*Taixu: A Study of the Modernization of Buddhism*). Taipei: Fagu wenhua 法鼓文化.

Jessup, J. Brooks, and Jan Kiely. 2016. *Recovering Buddhism in Modern China*. New York: Columbia University Press.

Jiang, Canteng 江燦騰. 1993. *Taixu dashi qianzhuan* 太虛大師前傳 (*Biography of Taixu's Early Years*). Taipei: Shin Wen Feng 新文豐.

Jones, Charles B. 2021. *Taixu's "On the Establishment of the Pure Land in the Human Realm": A Translation and Study*. London: Bloomsbury Academic.

Lai, Rongdao. 2017. The Wuchang Ideal: Buddhist Education and Identity Production in Republican China. *Studies in Chinese Religions* 3: 55–70. [CrossRef]

Lee, Amy. 2021. The Autobiographical Self of a Buddhist Monk: Brief Analysis of Master Yin Shun's *An Ordinary Life*. *Canadian Journal of Buddhist Studies* 16: 1–35.

Li, Shizhen 李時珍. 2021. *Ben Cao Gang Mu, Volume II: Waters, Fires, Soils, Metals, Jades, Stones, Minerals, Salts*. Translated by Paul U. L. Unschuld. California: University of California Press.

Li, Silong. 2013. The Practice of Buddhist Education in Modern China. *Chinese Studies in History* 46: 59–78.

Liu, Xuekai 劉學鍇. 2016. *Wen Tingyun quanji Jiaozhu* 溫庭筠全集校注 (*The Complete Collection of Wen Tingyun's Works with Annotations*). Taiyuan: Shanxi Publishing Media Group 山西出版傳媒集團.

Pittman, Don Alvin. 2001. *Toward a Modern Chinese Buddhism: Taixu's Reforms*. Honolulu: University of Hawai'i Press.

Ritzinger, Justin. 2017. *Anarchy in the Pure Land: Reinventing the Cult of Maitreya in Modern Chinese Buddhism*. New York: Oxford University Press.

Shi, Taixu 釋太虛. 2005. *Taixu dashi quanshu* 太虛大師全書 (*Collected Works of Master Taixu*). 35 vols. Beijing: China Religious Culture Publisher 宗教文化出版社.

Shi, Yinshun 釋印順. 2011. *Taixu dashi nianpu* 太虛大師年譜 (*Chronological Biography of Taixu*), revised ed. Beijing: Zhonghua Book Company 中華書局.

Tan, Guilin 譚桂林. 2010. Qingmo Minchu Zhongguo de Fojiao wenxue yu qimeng sichao 清末民初中國的佛教文學與啟蒙思潮 (Buddhist Literature and Enlightenment Intellectual Currents in Late Qing and Early Republican China). *Zhongguo Shehui Kexue* 中國社會科學 3: 158–71.

Tarocco, Francesca. 2005. *The Cultural Practices of Modern Chinese Buddhism: Attuning the Dharma*. New York: Routledge.

Tian, Xiaofei 田曉菲. 2005. Illusion and Illumination: A New Poetics of Seeing in Liang Dynasty Court Literature. *Harvard Journal of Asiatic Studies* 65: 7–56. Available online: http://www.jstor.org/stable/25066762 (accessed on 1 June 2005).

Travagnin, Stefania. 2017. Buddhist Education between Tradition, Modernity and Networks: Reconsidering the 'Revival' of Education for the Sangha in Twentieth-century China. *Studies in Chinese Religions* 3: 220–41. [CrossRef]

Welch, Holmes. 1968. *The Buddhist Revival in China*. Cambridge: Harvard University Press, pp. 51–71.

Wen, Hangsheng 文航生. 2011. *Sikong Shu shiji jiaozhu* 司空曙詩集校注 (*The Annotated Collection of Sikong Shu's Poetry*). Beijing: Zhonghua Book Company.

Xiang, Chu 項楚. 2000. *Hanshan shi zhu* 寒山詩注 (*Notes on the Poetry of Hanshan*). Beijing: Zhonghua Book Company.

Xie, Siwei 謝思炜. 2006. *Bai Juyi shiji jiao zhu* 白居易詩集校注 (*Notes on and Correction of Bai Juyi's Poetry Collection*). Beijing: Zhonghua Book Company.

Yan, Kejun 嚴可均. 1999a. *Quan Hanwen* 全漢文 (*The Complete Works of Han Dynasty Literature*). Beijing: The Commercial Press 商務印書館.

Yan, Kejun 嚴可均. 1999b. *Quan Jinwen* 全晉文 (*The Complete Works of Jin Dynasty Literature*). Beijing: The Commercial Press.

