# Peer review of "“Lamp and Candle”: Classical Chinese Imagery in Taixu’s Poetry"

_religions, doi:10.3390/rel14081077_

Round 1

Reviewer 1 Report

The manuscript provides an original and insightful exploration into the use of 'lamp and candle' imagery in Taixu's poetry, offering a unique perspective that contributes to the broader understanding of Taixu's work and the field of Buddhist literature. The depth of analysis is commendable, with the manuscript delving into the symbolic meanings of 'lamp and candle' imagery, connecting it to various aspects of Buddhism such as enlightenment, the fleeting nature of life, and the cultivation of wisdom. The use of primary sources is extensive, demonstrating a thorough engagement with Taixu's poetry. Furthermore, the manuscript successfully situates Taixu's use of 'lamp and candle' imagery within the broader context of his life and work, as well as the cultural and religious traditions that influenced him.

The main idea of the manuscript is that Taixu uses 'lamp and candle' imagery in his poetry as a symbol of various aspects of Buddhism, including enlightenment, the fleeting nature of life, and the cultivation of wisdom. This imagery is deeply rooted in Taixu's personal experiences and cultural background, and it serves as a powerful tool for expressing his religious beliefs and emotional states.

However, there are several areas where the manuscript could be improved.

1. **Lack of clarity in the use of metaphors:** The manuscript frequently uses the metaphor of a lamp or candle to symbolize various aspects of Buddhism, such as enlightenment and the fleeting nature of life. However, the metaphor is not always clearly explained or connected to the points being made. For example, on page 5, lines 175-178, the manuscript states, "The virtues of the lamp are comparable to those of the Buddha’s fruition, as the lamp dispels darkness, the profound compassion of the Buddha also dispels the ignorance of sentient beings." While the metaphor is poetic, it could be made clearer how exactly the virtues of a lamp are comparable to those of the Buddha's fruition. 

2. **Lack of context:** The manuscript often quotes from Taixu's poetry without providing sufficient context. For example, on page 7, lines 255, the manuscript quotes a line of Taixu's poetry: "We once shared the brilliance under the lamp’s glow, Today, revisiting, memories of you multiply and grow." However, the manuscript does not provide enough context about the situation in which this poem was written or the relationship between Taixu and the person he is addressing in the poem. 

3. **Potential bias:** The manuscript seems to present Taixu's views on Buddhism and his use of lamp and candle imagery in a largely positive light, without critically examining or questioning these views. For example, on page 2, lines 87-91, the manuscript describes Taixu's early experience with a lamp in a way that seems to romanticize it, without critically examining how this experience might have influenced his later views and writings.

4. **Lack of counterarguments or alternative interpretations:** The manuscript does not seem to present any counterarguments or alternative interpretations to Taixu's views or his use of lamp and candle imagery. This could limit the depth and balance of the analysis.

5. **Structure and organization:** The manuscript could benefit from a clearer structure and organization. For example, it could start by providing an overview of Taixu's life and work, then move on to a detailed analysis of his use of lamp and candle imagery, and finally discuss the implications of this imagery for understanding Taixu's views on Buddhism. 

Reviewer 2 Report

It is hard to find the research significant in this article. 

Firstly, the author does not elaborate on how this study could fill the current research gap. Simple to say, why so important to investigate Taixu's "lamp and candle" imagery? How does this study deepen our understanding of Taixu's Buddhist thought or modern Chinese Buddhism? 

The main text is long, but half of it is fulfilled by the text of the quote from Taixu's poetry. It is a lack of arguments, discussions and analysis. Does the "lamp and candle" has symbolism related to Humanistic Buddhism? Besides Taixu, do other Buddhist monks use the imagery? Any uniqueness of this "lamp and candle" imagery to modern Chinese Buddhism? 

The author focuses only on Taixu, but for research, it should be connected with a religious background. The research results seem to fail to engage with current Buddhism studies. 

The article is more Chinese written style. The author should smoothen the language. It is not necessary to quote so much poetry in the main text.    

Reviewer 3 Report

This is a an excellent study of Taixu's poetry. Your writing is very clear and your argument is convincingly presented. Your poetic translations are also very beautiful. Thank you for the excellent work.

Author Response

Thank you for reviewing this manuscript and providing your valuable feedback. I am grateful for your commendations on the writing, argument, and poetic translations presented in the article. Your support and scholarly input are greatly appreciated.

Round 2

Reviewer 1 Report

I am highly impressed by the way the authors have considered and addressed the previous comments and concerns in this revision. They have demonstrated professionalism and commitment to the advancement of knowledge in their respective field by incorporating the suggested modifications into their work. The changes have clearly enhanced the overall quality of the paper, resulting in a more coherent and comprehensive study.

Each concern raised in my initial review has been satisfactorily addressed, resulting in a significantly improved manuscript. Consequently, I believe this manuscript is now ready for publication and strongly recommend its acceptance.

Author Response

Thank you for your thoughtful and positive comments on my revised manuscript. I am delighted to hear that my efforts to address the previous concerns have been well-received and have contributed to enhancing the overall quality of the article.

Reviewer 2 Report

The author just did a minor modification to the article. The author should be elaborated and responded more to the questions I told in the first review. 

In addition, some of the citations could be put in the main text. The citations should be changed to "notes" and put the main text behind each page's bottom instead. 

Round 3

Reviewer 2 Report

Thank you for the hard work done by the author. I am satisfied with the article after the second amendment. There is no problem with the citations (just a reminder for the author to follow the article format).  

The article could be published in its current form. 

Author Response

Thank you for your positive evaluation of this revised manuscript.

I appreciate all your valuable remarks and suggestions, especially those related to Taixu's poetry's relationship with modern Chinese Buddhism. These insights have significantly improved and strengthened the manuscript.